# Evaluation of Existing Models to Estimate Sorption Coefficients for Ionisable Pharmaceuticals in Soils and Sludge

**DOI:** 10.3390/toxics8010013

**Published:** 2020-02-11

**Authors:** Laura J. Carter, John L. Wilkinson, Alistair B. A. Boxall

**Affiliations:** Department of Environment and Geography, University of York, York YO10 5NG, UK; john.wilkinson@york.ac.uk (J.L.W.); alistair.boxall@york.ac.uk (A.B.A.B.)

**Keywords:** hydrophobic partitioning, acid, base, uncharged, fate

## Abstract

In order to assess the environmental risk of a pharmaceutical, information is needed on the sorption of the compound to solids. Here we use a high-quality database of measured sorption coefficients, all determined following internationally recognised protocols, to evaluate models that have been proposed for estimating sorption of pharmaceuticals from chemical structure, some of which are already being used for environmental risk assessment and prioritization purposes. Our analyses demonstrate that octanol-water partition coefficient (*K_ow_*) alone is not an effective predictor of ionisable pharmaceutical sorption in soils. Polyparameter models based on pharmaceutical characteristics in combination with key soil properties, such as cation exchange capacity, increase model complexity but yield an improvement in the predictive capability of soil sorption models. Nevertheless, as the models included in this analysis were only able to predict a maximum of 71% and 67% of the sorption coefficients for the compounds to within one log unit of the corresponding measured value in soils and sludge, respectively, there is a need for new models to be developed to better predict the sorption of ionisable pharmaceuticals in soil and sludge systems. The variation in sorption coefficients, even for a single pharmaceutical across different solid types, makes this an inherently difficult task, and therefore requires a broad understanding of both chemical and sorbent properties driving the sorption process.

## 1. Introduction

In order to fully assess the environmental risk of a pharmaceutical active ingredient, information is needed on the sorption of the compound to environmental solids such as sewage sludge, soil, and sediment. Sorption partition coefficients (*K*_d_) are used to describe the extent to which an organic chemical is distributed, at equilibrium, between a solid and the aqueous phase that it is in contact with. Sorption coefficients, therefore, provide insights into the mobility and bioavailability of a compound in soil or sediment [1], as well as the extent of its removal from wastewater via sorption to sludge [2] and its propensity to associate with bed sediments [3]. A sorption coefficient is, therefore, a key component in the environmental risk assessment process of pharmaceuticals and is used in Phase I of the European Medicines Agency guideline for environmental risk assessment of pharmaceuticals to evaluate the requirement for soil and groundwater assessment and to estimate predicted environmental concentrations (PECs) in Phase II [4].

Sorption coefficients are known to vary widely for a given pharmaceutical active ingredient in different soil and sludge types [2,5,6,7]. This variation is thought to be due to a range of drivers, including differences in the following: quantity of organic matter (OM); particle size distributions and surface area of the sorbent; the pH, ionic strength, and the dissolved organic matter concentration of the solution; and concentrations of solids used in the studies [6,8,9,10,11]. In particular, studies have highlighted the importance of pH in explaining variations in the sorption of pharmaceuticals in different systems. With an estimated >60% of pharmaceuticals containing an ionisable group [12] and environmental pH known to vary by up to 7.5 pH units (e.g., pH in surface waters in Europe is reported to range from 2.2 to 9.8), pharmaceuticals can dissociate to different degrees in the environment which can alter the fate and behaviour of the chemical [13,14,15,16,17]. It is, therefore, essential that effects of variations in environmental pH on sorption are recognised when assessing the environmental risks of a pharmaceutical.

Differences in the proportions of ionised and nonionised species can influence specific sorption mechanisms and ultimately lead to differences in the strength of sorption at different pH values. The nonionised form of a molecule most likely associates with the organic carbon (OC) of the solid through hydrophobic interactions [18,19]. Anionic species form complexes with divalent and trivalent cations which then sorb to the negative surfaces on particles via electrostatic attraction or through surface bridging mechanisms [20,21]. Interaction with positively charged surface oxides has also been proposed as a mechanism for sorption of anionic species [22,23]. Cationic species can be electrostatically attracted to negatively charged environmental substrates. Specifically, organic matter and phyllosilicate clay minerals contribute to the sorption affinity of organic cations where cation exchange processes occur at negatively charged surface groups [8,9,24]. The relative importance of one mechanism over another depends on the solid constituents, the molecule, and the characteristics of the environment in which the solid resides.

Sorption coefficients (*K*_d_) can be obtained by experimental determination, for example, using the Organisation for Economic Co-operation and Development (OECD) test guideline 106 [25]. However, given that >1500 pharmaceuticals are currently marketed [26], with others continuously being developed for market, and the large variability in the characteristics of soils, sediment, and sludge across the natural environment, it would be beneficial to have quantitative structure-property relationships (QSPRs) that are able to estimate the sorption behaviour of a pharmaceutical to soils, sediments, and sludges with varying characteristics. Such models could be combined with spatial information on soil and sediment properties to better understand the fate and transport of an active substance across a landscape.

While a large number of QSPRs are available for estimating the sorption behaviour of nonionised organic compounds to soils, sediments, and sludges, fewer models are available for ionisable chemicals which also include terms related to the matrix properties (e.g., soil pH) [27]. Examples of models for soils include regression-based QSPRs for acids and bases [24,28,29], and models based on the summed contribution of sorption to key soil properties [30], and pH-dependent speciation [16,31]. For sludge, available QSPRs that account for ionisation include regression-based models such as those developed by Sathyamoorthy and Ramsburg [32] and Franco et al. [33], as well as Bayesian artificial neural networks (ANNs) such as those proposed by Berthod et al. [34] and Barron et al. [35].

While each of these models has been evaluated to some degree, the level of evaluation is limited due to the availability of experimentally derived data on the sorption behaviour of pharmaceuticals in the environment that is in the public domain. The quality of some data in the public literature is also questionable with a lack of consistency in experimental conditions and studies neglecting to utilise validated analytical techniques and standardised guidelines. As part of the Innovative Medicines Initiative project ”Intelligent Led Assessment of Pharmaceuticals in the Environment” (iPiE), thirteen pharmaceutical companies that are members of the European Federation of Pharmaceuticals Industry Associations (EFPIA) have been working with the research sector to develop a unique database containing results of regulatory environmental fate and effects studies that have been performed on pharmaceuticals. A large component of this data is available in the iPiE*SUM component of the database (https://ipiesum.eu/) which is accessible to the research community. This database includes previously unpublished data to support the regulatory risk assessment of pharmaceuticals, generated using Good Laboratory Practice (GLP), a set of principles intended to assure the quality and integrity of laboratory studies. This unique, high quality database encompasses data on the sorption of pharmaceuticals in soils and sludge providing an opportunity to more thoroughly evaluate sorption models proposed in the literature. Therefore, in this study, we use the sorption data from the iPiE database to evaluate the performance of existing models that have been proposed for estimating the sorption of ionisable pharmaceuticals. Models were selected that account for chemical speciation by incorporating terms related to matrix properties (e.g., soil pH and lipophilicity corrected for pH (log *D*)). Whilst additional sorption models have been published that specifically account for the presence of ionisable functional groups and are based on mechanistic understanding of sorption to soils, these models often require detailed soil properties which are not commonly reported in standard sorption studies (e.g., ter Laak [29]) or experimentally determined sorption coefficients of a probe compound in the soil of interest (e.g., Jolin et al. [36]). It was, therefore, not possible to evaluate all existing sorption models as input data needed for these models was not available in the iPiE database or could not be computed experimentally.

## 2. Materials and Methods

Data donation from EFPIA companies enabled the generation of a dataset of high-quality sorption coefficients and associated metadata for 83 pharmaceuticals (with octanol-water partition coefficients (log *K*_ow_) ranging from −4.13 to 7.59) in a variety of soil types and 58 pharmaceuticals (log *K*_ow_ ranging from −2.13 to 7.13) in sludge. All studies were conducted according to GLP and were based on the OECD test guideline 106 [25] or United States Food and Drug Administration (US FDA) Technical Assistance Document 3.08. These protocols require preliminary identification of the most suitable soil/solution ratios to ensure that equilibrium of the test compound between the soil and soil solution is reached by the end of the experiment. The sorption coefficients are subsequently calculated using adsorption isotherm experiments across five concentrations, covering two orders of magnitude. Mathematical sorption models (e.g., Linear or Freundlich adsorption isotherm) are, then, used to describe the sorption process and calculate a sorption coefficient. The molecules for which the *K*_d_ values were available were separated into acids, bases, and chemicals which have multiple ionisable functional groups based on the ability of the chemical to accept or donate hydrogens, following the Bronsted–Lowry definition of acids and bases. The degree of ionisation of the chemical at the experimental pH was then calculated.

In total, 287 soil sorption coefficients (and associated data on soil and sludge properties) for 83 pharmaceuticals were provided by EFPIA companies, with data being available for 21 chemicals containing an acidic functional group, 41 with a basic functional group, and 21 containing both acidic and basic functional groups. For sludge, 84 sorption coefficients were available for 58 different pharmaceuticals, comprising of 20 chemicals containing an acidic functional group, 25 with a basic functional group, and 13 containing both acidic and basic functional groups.

Log sorption coefficients (log *K*_d_) for the compounds in the dataset ranged between −3.70 to 4.56 and −0.7 to 7.4 L/kg for soil and sludge, respectively. For each pharmaceutical where soil sorption data were available, sorption coefficients were typically determined in 3 to 5 different soil types with a wide range of properties, including pH (ranging from 4 to 8.2), OC content (ranging from 0.5% to 17%), and cation exchange capacities (CEC) (ranging from 0.35 to 52.5 meq/100g). For the sludge data, pH values for the test systems ranged from 5.2 to 7.8 and the OC content ranged from 20.1% to 89.4%. For a detailed list of chemical properties of each test compound, as well as soil and sludge parameters, see Appendix A. Pharmaceuticals have been anonymised as some of this data is not currently publicly available.

### 2.1. Evaluated Models

Models were selected that account for chemical speciation by incorporating terms related to matrix properties (e.g., soil pH and log *D*). Models for soil and sludge sorption are presented in Table 1 and Table 2, respectively. For soils, models evaluated included the approach of Bintein and Devillers [28] who proposed a general QSAR model that uses the physicochemical properties of the molecules (i.e., log *K*_ow_) together with some relevant soils properties (i.e., pH and %OC) to estimate the sorption behavior of both ionised and nonionised chemicals. The regression equation, including Log *D* and the soil OC content to predict the adsorption of acids by Kah and Brown [24], was also selected for evaluation. Predictions from a regression equation proposed by Franco and Trapp [31] for acids accounting for both the neutral and ionic molecule species together with a nonlinear regression equation for bases were compared to measured data in our database. Moving away from a regression equation, we evaluated the quantitative model for organic cations proposed by Droge and Goss [30] which was based on an overall *K*_d_ calculated from the summed contribution of sorption to organic matter and sorption to phyllosilicate clay minerals. Although the *K*_ow_ based regression QSPR developed by Sabljić et al. [37] was not specifically developed for ionisable chemicals, this model was selected given its importance in currently accepted risk assessment practices (EU Technical Guidance Document [38]) and the potential for it to be used for pharmaceuticals which are ionised at environmental pH.

Models for sludge sorption included approaches by Franco et al. [33], who proposed regressions for monovalent acids and bases based on *K*_ow_, the dissociation constant (p*K*a), and the pH. Models by Sathyamoorthy and Ramsburg [32], including single and polyparameter QSPRs, were also evaluated together with the ANN proposed by Berthod et al. [34]. The specified range of chemical applicability and training dataset for each model are summarised in Table 1 and Table 2. It should be noted that the evaluation of the models is, therefore, limited to chemicals in our test dataset that fall within the applicability domain of the selected model. For example, published models are in most cases only suitable for chemicals with single ionisable functional groups, such as Franco et al. [16], and therefore zwitterionic compounds were often removed.

#### Generation of Chemical Descriptors for Inclusion in the Models

A variety of descriptors was needed to estimate sorption behaviour using the different existing QSPR models. Log *K_ow_* and p*Ka* values were estimated using the ACD/I-Lab software (v. 2018). The number of hydrogen bond acceptors (nHBA), log of molecular volume (log MV), and log of topological surface area (log TPSA) were calculated using the Molinspiration online interface calculator (http://www.molinspiration.com/) as detailed by Sathyamoorthy and Ramsburg [32]. Energy minimised, three-dimensional conformations for input into the ANN were generated from canonical SMILES using the Molecular Operating Environment (MOE) (2015.10), according to methods outlined by Berthod et al. [34]. Relevant values were then input into the models to estimate either the *K*_d_ or *K*_oc_ of a molecule. For consistency, in instances where the QSPR estimated the *K*_oc_ of a molecule, the *K*_oc_ was converted to a *K*_d_ using information on the soil organic carbon content before predictions were compared with the measured *K*_d_ values. Results of *K*_oc_ regression analysis prior to conversion to *K*_d_ are provided in the Appendix A.

### 2.2. Statistical Analysis

The performance of the published models was assessed using a suite of statistical tests using SPSS (v. 24). The ability of each model to capture the variance in the dataset used to develop the model was evaluated using the correlation coefficient (*r*^2^). The predictive capability of models was assessed using the Nash–Sutcliffe efficiency (*NSE*) (Nash and Sutcliffe 1970). The *NSE* ranges from −∞ to 1. Negative NSE values indicate that the mean of the measured values is a better predictor than the model. Absolute errors were also calculated (root mean square error (RMSE) and mean absolute error (MAE)) to measure the average magnitude of the error in the prediction. A ratio of RMSE and MAE was used as an indicator of the extent to which outliers affect the model evaluation, with a value >1 suggesting there are outliers and RMSE = MAE suggesting that all errors are of the same order of magnitude. The percent of predictions within a factor of 10 of the measured sorption coefficients was also calculated. This was considered to be an acceptable threshold for a prediction, as this level of variability could be expected in experimental results between studies carried out on the same solid-liquid system in different laboratories.

## 3. Results

Measurements of sorption coefficients are time and resource consuming and predictive approaches are, therefore, an attractive alternative to generate these data. In this study we therefore evaluated nine soil and five sludge models with the goodness of fit parameters provided in Table 3 and Table 4. Regressions of measured sorption coefficients against predicted sorption coefficients are provided in Figure 1, Figure 2, Figure 3 and Figure 4. A number of sorption coefficients were removed from the analysis where chemicals fell outside of the specified applicability domain of the respective models (see Table 1 and Table 2 for model applicability domains). For example, zwitterions and amphoteric chemicals were removed from the analysis unless the model explicitly stated that it could be used for chemicals containing more than one ionisable functional group (e.g., Berthod et al. [34]). Pharmaceuticals within the applicability domain, and included in the analysis of each model, are provided in Appendix A. An evaluation of the ratio between RMSE and MAE would suggest that outliers were, on the whole, not affecting the model performance for soil and sludge models as all values were less than one (Table 3 and Table 4).

### 3.1. Soil Sorption

In total, nine sorption models proposed for use in the environmental fate assessment of ionisable chemicals were evaluated for accuracy in the prediction of ionisable pharmaceutical sorption to soils (Table 3). The hydrophobic chemical QSAR, proposed by Sabljić et al. [37], was not developed to predict the sorption coefficients of ionisable chemicals specifically, yet was able to predict 55% of the measured *K*_d_ values to within a factor of 10 (Figure 1). Log *K*_d_ values predicted within 1 log unit included acids and bases which were charged at experiment soil pH, and therefore displayed anionic (10 negatively charged acids) and cationic (26 positively charged bases) properties. However, the accompanying statistics demonstrated poor model performance with low predictive power (*r*^2^ 0.07, *NSE* −1.94) (Table 3). A significant deviation from the 1:1 line was observed, and measured log *K*_d_ values were generally underpredicted by this QSAR (Figure 1). The organic acids QSAR by Sabljić et al. [37] similarly led to an underestimation of sorption coefficients for the acidic pharmaceuticals included in our dataset (Figure 2) resulting in a poor model performance (*r*^2^ = 0.13) with low predictive capability (RMSE 0.63, *NSE* −0.82) (Table 3).

The lowest *r*^2^ and greatest negative *NSE* value was calculated for the Kah and Brown approach [24] (*r*^2^ 0.003, *NSE* −7.71) for the prediction of sorption coefficients for acids. Surprisingly, this model resulted in the largest proportion of *K*_d_ values predicted within a factor of 10 of the corresponding measured value (71%, Table 3). However, due to the limited training set of chemicals used to develop this model (1.97 ≤ p*K*a ≤ 4.94), only seven chemicals were within the applicability domain of this model (Figure 2, Table 3). Limited applicability of this model combined with statistics demonstrating poor predictive capability suggest that this approach is limited for the prediction of sorption coefficients for acidic pharmaceuticals.

Models specifically developed for acids and bases by Bintein and Devillers [28] also performed poorly, and resulted in a large amount of variation around the 1:1 line with predictions typically underestimated (Figure 2 and Figure 3, Table 3). A smaller number of predictions were within a factor of 10 of the corresponding measured sorption coefficient (*K*_d_) for acids (26%) and bases (24%) than those observed for the Sabljić model [37] (>50%) (Table 3). The Bintein and Devillers approach [28] also resulted in very large negative *NSE* indexes for acids (−4.43) and bases (−9.65) (Table 3). In these cases, the analysis indicates that the mean of the measured *K_d_* values is a better predictor than the models included in this analysis, and in general, model performance was low.

The Franco et al. models [16,31] also resulted in negative *NSE* values but were in the range of –0.26 to −0.31 which were smaller than calculated for the approaches by Bintein and Devillers [28], Kah and Brown [24], and Sabljić et al. [37] (Table 3). Considering soil sorption as a summed contribution of species-specific sorption coefficients [16,31] resulted in 55% to 68% of the predicted *K*_d_ values being within a factor of 10 of the corresponding measured value. However, there was still a substantial amount of variation around the 1:1 line (Figure 2, RMSE values >0.65) and both the Franco and Trapp [31] and Franco et al. [16] approaches underestimated the log *K*_d_ of highly hydrophobic acidic compounds with log *K*_d_ values >2. With the exception of a small number of chemicals, the bases model proposed by Franco and Trapp [31] was observed to generally underestimate sorption coefficients by up to a factor of two (Figure 3). Nevertheless, the approach presented by Franco and Trapp, in 2008, for monovalent acidic compounds [31], with the later adjustment of including a term to account for optimum soil pH in 2009 [16], was the best performing model to predict sorption of soils for pharmaceuticals that contain a single acidic functional group.

Comparatively, the Droge and Goss model [30] resulted in the best model performance for the prediction of sorption coefficients for pharmaceuticals with a basic functional group. Comparison of predicted and measured sorption coefficients resulted in a relatively small deviation from the 1:1 line and 71% of the predicted log *K_d_* values were within one log unit of the corresponding measured log *K_d_* value (Figure 3 and Table 3). There is a larger spread of data around the 1:1 line for pharmaceuticals from the iPiE database as compared with the dataset used to evaluate the model in the original paper, where it was observed that most *K*_d_ predictions were within a factor of three [30]. The Droge and Goss model [30] was also able to capture a larger proportion of the variance in the measured data with a relatively high *r*^2^ of 0.29. However, a large RMSE of 0.79 (Table 3) suggests that there is a need to improve this prediction further and reduce the associated standard deviation of unexplained variance.

Even though all models evaluated here resulted in low *r*^2^ values (<0.29, Table 3), the results show that approaches that consider the charge of the chemical in combination with information on selected soil properties perform better than the *K_ow_*-based models. This finding is in agreement with previous studies which have concluded that, based on advances in our mechanistic understanding, multiple reactive sites, in addition to organic matter, are important for ionisable chemical sorption [11,39]. For example, oxytetracycline can sorb to soils via a number of mechanisms including cation exchange and cation bridging to negatively charged sites on both aluminosilicates and organic matter, as well as complexation to iron and aluminium ions on metal oxides. Models that also included pharmaceuticals in their training sets for model development also performed better than models that were developed for general group organic chemicals.

### 3.2. Sludge Sorption

The Franco approach of considering sorption as the sum of sorption from the nonionised fraction and the ionised fraction also appears to work well for predicting sludge sorption of acids, using a similar model to that published for soils (Table 4). The model proposed by Franco et al. [33] for monovalent bases was able to predict the largest percent of measured *K_d_* values to within one log unit (67%) (Table 4 and Figure 4). However, further statistical evaluation of the model suggested poor model performance with an *NSE* of −0.07 and *r*^2^ of only 0.04. Model estimates for monovalent acids [33] were within one order of magnitude for 60% of the sorption coefficients included in the analysis but similarly to the regression proposed for bases, model predictions were poor and resulted in a large mean deviation from the measured sorption coefficients (RMSE 0.89, *r*^2^ 0.07) (Table 4 and Figure 4). The polyparameter QSAR’s proposed by Sathyamoorthy and Ramsburg [32] for acids also resulted in poor predictions of log *K_d_* (RMSE 0.31 to 0.32, *NSE* (−9.28) to (−12.68), *r*^2^ 0.04–0.08) (Table 4 and Figure 4). Even though these QSAR’s were developed using published sorption coefficients for pharmaceuticals specifically, our analysis resulted in a poorer model performance to that reported in the original manuscript (*NSE* 0.61–0.64, *r*^2^ 0.60–0.64) [32]. Variation in model predictions from the ANN by Berthod et al. [34] was relatively small (*r*^2^ 0.21), however all model predictions were a factor of approximately two to three larger than the measured values which resulted in a large deviation from the 1:1 line (*NSE* index −2.38) and only 21% of predicted *K_d_* values within a factor of 10 of the corresponding measured value (Table 4). Both the models proposed by Berthod et al. [34] and Sathyamoorthy and Ramsburg [32] were developed using pharmaceutical sorption coefficients published in the scientific literature. Given both these models were consistent in either under- or overpredicting sorption coefficients in the iPiE database, this would suggest that there are some common, but significant differences in sorption coefficients calculated following internationally accepted guidelines for regulatory risk assessment and those determined primarily for research purposes.

As the model proposed by Berthod et al. [34] was not charge specific, all chemicals, regardless of their percent ionisation, were included in the analysis. The model was evaluated further by separating the chemicals and associated *K_d_* values into the three groups (acids, bases, and chemicals containing multiple ionisable functional groups) and repeating the analysis (Table 4). Whilst 30% chemicals classed as either amphoteric or zwitterionic (multiple ionisable functional groups) were predicted within a factor of 10 of the corresponding measured value (*K*_d_), the accompanying statistics suggest the ANN has a poor predictive capability for this group of chemicals (*r*^2^ 0.01, *NSE* −17.50, Table 4) as compared with acids and bases. Analysis of the acid and base *K_d_* values separately resulted in comparable statistics to when the data was evaluated as a whole, with 19% of predictions within a factor of 10 of the measured value (Table 4).

## 4. Discussion

### 4.1. Deviation from Neutrality

Broadly defining a model as suitable for acidic or basic chemicals means that some chemicals, within the applicability domain of the model, have functional groups that are ionised at test pH (e.g., amine p*K*a > 8) and other chemicals remain relatively un-ionised. The presence or absence of ionisable functional groups can strongly influence the sorption of a pharmaceutical to soils and sludge, and therefore models developed to predict ionisable chemical sorption need to account for this. Sathyamoorthy and Ramsburg [32] determined that the predictive capability of models based on chemical hydrophobicity only became meaningful when the percentage of uncharged chemical was >99%. However, models considered in this analysis typically show a larger ratio between predicted and measured log *K*_d_ values for soils and sludge when the pharmaceutical is not displaying an acidic or basic charge and is >99% uncharged (Figure 5 and Figure 6). For soils, this finding is most interesting with regards to the Sabljić hydrophobics model [37] which is often used as a default to predict the sorption of nonionised pharmaceuticals to soil. Our analysis shows that this model, and similarly the Sabljić acids model [37], perform the worst for pharmaceuticals which were completely un-ionised at test soil pH (Figure 5) with the ratio ranging between −15 and 23 for predicted log *K*_d_ value to measured log *K*_d_ values. A large deviation between predicted and measured soil sorption coefficients were also observed for the Franco suite of models [16,31] when the chemical was classed as uncharged (Figure 5). The ratios of predicted and measured sludge sorption coefficients for uncharged chemicals were, however, smaller than ratios calculated for soils, with all ratios within a factor 10 of the 1:1 line (Figure 6).

Comparatively, when a chemical is highly charged (>90% ionised) this also results in a larger ratio between predicted and measured log *K*_d_ values (Figure 5). The Bintein and Devillers [28] model analysis clearly shows that for both acids and bases, as the percent ionisation increases this generally results in a larger deviation between the predicted and measured soil log *K*_d_ values (Figure 5). For acids, this typically results in an underestimation of the sorption coefficients, with 80% of sorption coefficients underestimated when the chemical was >90% ionised at test soil pH. Likewise, the bases model appears to work well for chemicals that were not significantly ionised, however, 90% of soil sorption coefficients were over predicted for basic pharmaceuticals which were >90% ionised (Figure 5). While the Droge and Goss model [30] appears to work well in the prediction of sorption coefficients for charged basic compounds (Figure 3 and Table 3), further analysis shows that this model does not perform well for basic pharmaceuticals that are highly ionised (>90%) as compared with moderately ionised bases (Figure 5). In their discussion, Droge and Goss [30] noted that this model was developed for strong bases (p*K*a > 8) which would be largely protonated at soil pH and the model applicability for neutral bases (p*K*a 3 to 8) was unknown. All chemicals tested in this analysis fell within the specified applicability domain (p*K*a > 8) but our analysis would indicate that this model would also work well for weak bases given that there is a smaller difference between measured and predicted sorption coefficients with a decreasing percent of ionisation (Figure 5).

### 4.2. Regulatory Implications

In a regulatory context, sorption models have been proposed for the environmental risk assessment of organic chemicals. Whilst the Sabljić hydrophobics model [37] was not specifically designed to predict ionisable chemical sorption, it is included in the Technical Guidance Document (TGD) [38] as one of a suite of suggested models to predict the partitioning of organic chemicals in soils. The results from this analysis (Table 3), including the inability of the Sabljić hydrophobics model [37] to capture the proportion of variance observed in the measured data (*r*^2^ of 0.07) and the low predictive power of the model (*NSE* −1.54) demonstrate that it should not be used in the risk assessment of ionisable pharmaceuticals and alternative models should be proposed to provide a better indication of soil sorption potential. Even the class specific TGD QSAR for organic acids (Table 3), resulted in poor model performance with only 50% of the predicted sorption coefficients being within an order of magnitude of their measured value (*r*^2^ 0.04, *NSE* −1.25).

In the risk assessment of pharmaceuticals, it is also desirable to have accurate methods to predict the sorption of pharmaceuticals to sewage sludge, as this is a key input into exposure models and the prediction of surface water concentrations. Without accurate information on partitioning and removal from the waste stream during wastewater treatment, predicted environmental concentrations will be under- or overestimated [40]. ”SimpleTreat” is an exposure emission model and originally included approaches to describe sludge sorption that do not account for ionisation but has since been updated to include QSARs to estimate ionisable chemical fate in an activated sludge WWTP [41]. Results from this model evaluation suggest the new QSARs (Table 4) [33] are able to predict sludge sorption coefficients reasonably well for acids and bases with the largest percent of *K*_d_ values being within a factor of 10 of the measured value obtained using these models (Figure 2).

Models that appear to be more applicable to ionisable organic compounds, such as nonlinear regressions, or those based on molecular structure, tend to be somewhat more complex in nature, and therefore can lack the transparency that the linear regression-based approaches provide. From a regulatory perspective, methods that tend to be simple and transparent are preferable to enable ease of use and straightforward implementation in risk assessment frameworks. However, based on our evaluation, such methods do not appear to perform well for ionisable organic compounds, typically underestimating the sorption coefficients (Figure 1 and Figure 2). Risk assessment practices for estimating sorption need to be updated to account for the complexities involved in pharmaceutical–sorbent interactions, especially with regards to current accepted practices for estimating soil sorption in the TGD [38].

### 4.3. Future Model Development

As polarity, number of functional groups, and ionic nature of the pharmaceutical increase, so does the number of potential sorption mechanisms, thereby resulting in differing sorption coefficients between nonionised and ionised chemicals. *K_ow_*-based QSARs do not account for the range of mechanisms involved in the sorption of ionisable chemicals when you assume hydrophobic partitioning is driving the sorption process and our analysis reflect this [42] (Figure 1). Similar findings were also observed by Sathyamoorthy and Ramsburg [32] who noted that single parameter models employing log *K_ow_* as the only descriptor performed poorly in their ability to predict pharmaceutical sorption to sludge. Interestingly, our analysis also demonstrated that models developed for ionisable chemicals specifically, and which included pharmaceuticals in their training sets, also resulted in significant differences between predicted and measured log *K*_d_ values for chemicals in the iPiE sorption database. New models, therefore, need to be developed that are able to capture the variance in sorption coefficients associated with different charged states of the chemical and properties of the soils and sludges.

A more detailed mechanistic understanding of soil and sludge sorption processes and specifically the key physicochemical properties responsible for sorbent–sorbate interactions could provide a strong basis for future model development. The approach proposed by Droge and Goss [30] is built on these principles by accounting for sorption of charged bases to organic matter and phyllosilicate clays, and yielded promising results in our analysis (Table 3). However, the prediction of *D*_OC,IE_ and *K*_CEC,CLAYS_ necessary for the overall sorption prediction using this model is based on simple amine structures and pharmaceuticals are typically comprised of more complex structures with specific chemical moieties. The availability of only a small set of poorly validated corrective increments to estimate *D*_OC,IE_ and *K*_CEC,CLAYS_ suggest these need to be updated and extended with new data to cover more complex multifunctional structures that are common in ionisable pharmaceuticals. Building on the work of Droge and Goss [30], Jolin et al. [36], demonstrated that you can predict organic cation sorption using a simple probe molecule, phenyltrimethylammonium, in combination with structural scaling factors derived from the Droge and Goss [30] approach. This hybrid approach offers promising advances in mechanistic derived sorption coefficients for organic cations, however, the need for an experimentally determined *K*_d_ value for the probe molecule in your soil type is a limiting factor in the Jolin et al. [36] approach.

The development of more mechanistically driven sorption predictions will also require a change in the standard suite of soil parameters that are reported for a sorption experiment. Ter Laak et al. [29] employed partial least squares-regression modelling to explain 68% to 78% of the variation in sorption coefficients for ionisable antimicrobial agents by integrating six different soil properties. Based on our current understanding of anionic sorption in soils, weak acids can strongly interact with positively charged surface oxides [22,23] and therefore inclusion of parameters such as this in QSAR models is logical. This approach, however, could not be included in our analysis as oxyhydroxide content, a key soil parameter required for these predictions, is not typically reported following OECD 106 or the US FDA Technical Assistance Document 3.08 protocols (the sources of data obtained for our analysis). To facilitate the use of this model, and the development of future models following a similar mechanistic approach, an increase in the types of soil parameters that are reported for a sorption experiment is required. Furthermore, it may be necessary to investigate a wider range of chemical properties and soil parameters in order to prioritise their ability to predict sorption. It is expected that sorbent properties such as oxide content and cation exchange capacity which are strongly correlated to the soil sorption of ionisable chemicals [8,20] will be less important in terms of describing sludge sorption. Sludge is a complex matrix but is largely dominated by a high organic matter content, which will be driving the intermolecular forces, primarily hydrophobic interactions in the sorption of nonionised and cationic chemicals in particular [13].

The use of ANN models could also provide a promising alternative to traditional regression-based approaches for sorption predictions by accounting for nonlinear relationships between a wide range of factors. Currently the suitability of ANNs to describe soil sorption is unknown. Future work could explore predictions of soil sorption coefficients based on ANN models, and incorporate terms related to soil properties (i.e., pH and OC%), in combination with pharmaceutical physicochemical parameters. Evaluation of the sludge ANN model developed by Berthod et al. [34] for sludge sorption revealed there was room for improvement in model predictions as it had weak predictive capability across acidic, basic, and nonionised pharmaceuticals (Table 4). As the Berthod et al. model [34] considered all chemicals, irrespective of charge in a single dataset, future sludge models based on ANN principles could be expanded by defining separate ANN models for acidic, basic, and nonionised pharmaceuticals.

The development of charge-based models (e.g., Franco et al. [31]) for acids and bases has started to account for different factors influencing the sorption of ionised chemicals. However, as our analysis demonstrates, some of the models developed for ionisable chemicals are still unable to accurately predict sorption coefficients for chemicals which are fully protonated (Figure 5 and Figure 6). Future model development, therefore, should take into account the degree of charge to improve predictions of sorption coefficients with separate models developed for pharmaceuticals which are highly protonated (>90% ionised).

Pharmaceuticals comprise a diverse array of chemical structures with a range of chemical moieties that can become charged at an environmentally relevant pH range. Previous research has elucidated that the extent of charge delocalisation (or positive charge localisation) and presence or absence of hydroxyl groups can strongly influence molecular orientation, and thereby the potential for ionised chemical sorption [43,44,45]. Therefore, this suggests that a single model to describe sorption accurately for all pharmaceutical classes may not be possible. Weber et al. [46] were unable to define a single equation that was able to explain the variation associated with all ionisable pesticides and, instead, proposed a series of mathematical equations from regressions of sorption coefficient values with selected soil properties for individual pesticide families (e.g., weakly basic compounds and carboxy acids). Due to the inherent complexity of compound structures, generalised sorption models are not be able to adequately describe the broad range of interactions involved in ionisable pharmaceutical sorption and models developed on the basis of structural similarity are required. A similar approach is also needed for other chemicals which are also ionised at environmental pH values (e.g., UV-filters and personal care products).

## 5. Conclusions

Using a high-quality database of measured sorption coefficients, all determined following internationally recognised protocols, our analyses demonstrate that *K_ow_* is not an effective predictor of ionisable pharmaceutical sorption in soils. This is in agreement with previously published studies. Polyparameter models based on pharmaceutical characteristics, in combination with key soil properties, such as cation exchange capacity, increased model complexity but yielded an improvement in the predictive capability of soil sorption models. Similar findings were observed with the sludge sorption models evaluated in this analysis, where charge-specific models accounting for chemical speciation provided a better prediction of sorption coefficients.

Our analysis has demonstrated that for a majority of pharmaceuticals, which have previously undergone an environmental risk assessment, existing models that have been proposed for estimating the sorption behaviour of ionisable compounds, fail to provide an accurate prediction for the partitioning to sewage sludge and soils. As the models included in this analyses were only able to predict a maximum of 71% and 67% of the sorption coefficients for the compounds to within one log unit of the corresponding measured value in soils and sludge, respectively, this would suggest there is a need for models to be developed to better predict the sorption of ionisable pharmaceuticals in soil and sludge systems. The variation in sorption coefficients, even for a single pharmaceutical, makes this an inherently difficult task, and therefore requires a broad understanding of both chemical and sorbent properties driving the sorption process to create mechanistic models that are able to respond to the variability in exposures. The model proposed by Droge and Goss [30] holds promise in this respect by characterising soils in a manner that parallels our mechanistic understanding, however, these are only limited to cations. The proposed ideas for future model development discussed above, offer steps towards improving upon current model uncertainty, and account for the broad spectrum of ionisable pharmaceuticals in the environment.

## Figures and Tables

**Figure 1 toxics-08-00013-f001:**
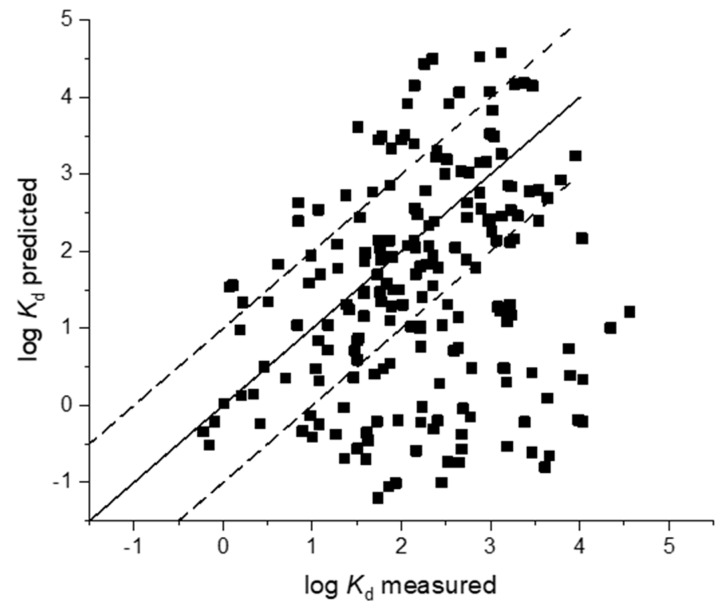
Comparison of predicted and measured soil sorption coefficients (log *K*_d_) using Sabljic et al. [37] model for hydrophobic chemicals.

**Figure 2 toxics-08-00013-f002:**
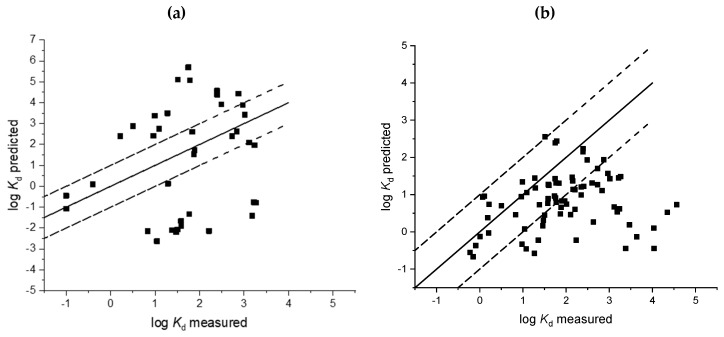
Comparison of predicted and measured soil sorption coefficients (log *K*_d_) for acidic pharmaceuticals Bintein and Devillers [28] (**a**); Sabljic et al. [37] (**b**); Franco et al. [31] (**c**); Franco et al. [16] (**d**); Kah and Brown [24] (**e**).

**Figure 3 toxics-08-00013-f003:**
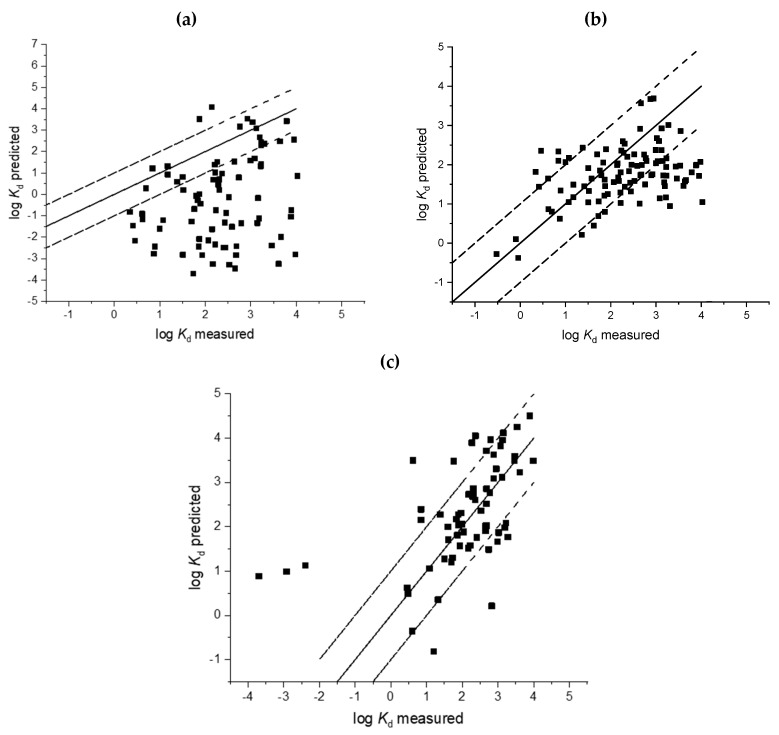
Comparison of predicted and measured soil sorption coefficients (log *K*_d_) for basic pharmaceuticals Bintein and Devillers [28] (**a**); Franco et al. [31] (**b**); Droge and Goss [30] (**c**).

**Figure 4 toxics-08-00013-f004:**
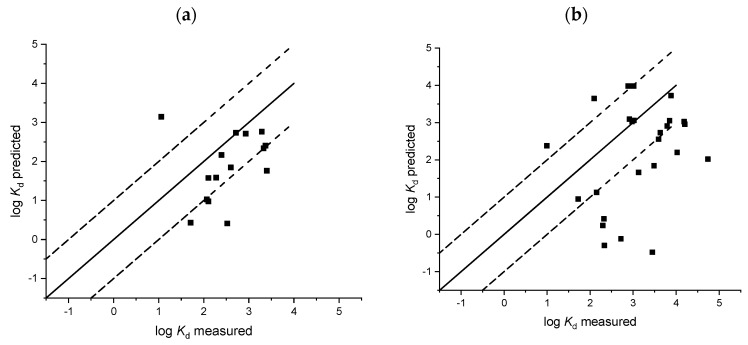
Comparison of predicted and measured sludge sorption coefficients (log *K*_d_), Franco et al. [33] acids (**a**); Franco et al. [33] bases (**b**); Sathymoorthy and Ramsburg [32] ionised acids (**c**); Sathymoorthy and Ramsburg [32] acids (**d**); Berthod et al. [34] (**e**).

**Figure 5 toxics-08-00013-f005:**
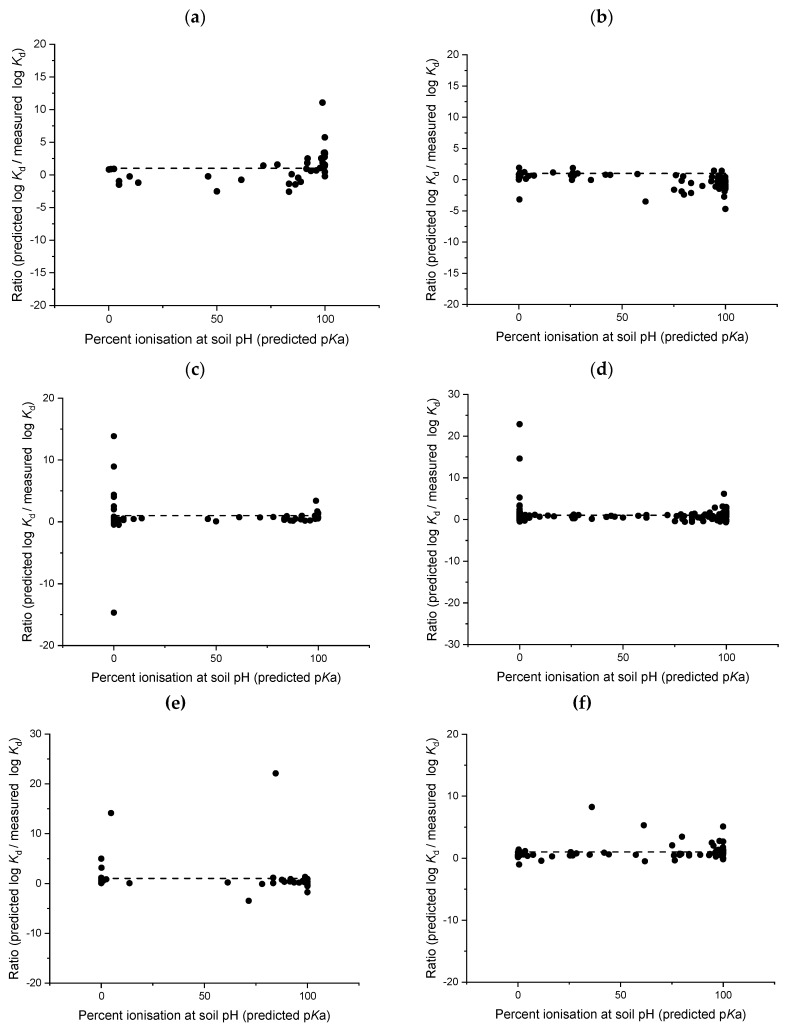
Ratio between predicted and measured soil sorption coefficients and relationship to percent ionisation of test compound, Bintein and Devillers [28] acids (**a**); Bintein and Devillers [28] bases (**b**); Sabljic et al. [37] acids (**c**); Sabljic et al. [37] hydrophobics (**d**); Franco et al. [31] acids (**e**); Franco et al. [31] bases (**f**); Franco et al. [16] (**g**); Kah and Brown [24] (**h**); Droge and Goss [30] (**i**).

**Figure 6 toxics-08-00013-f006:**
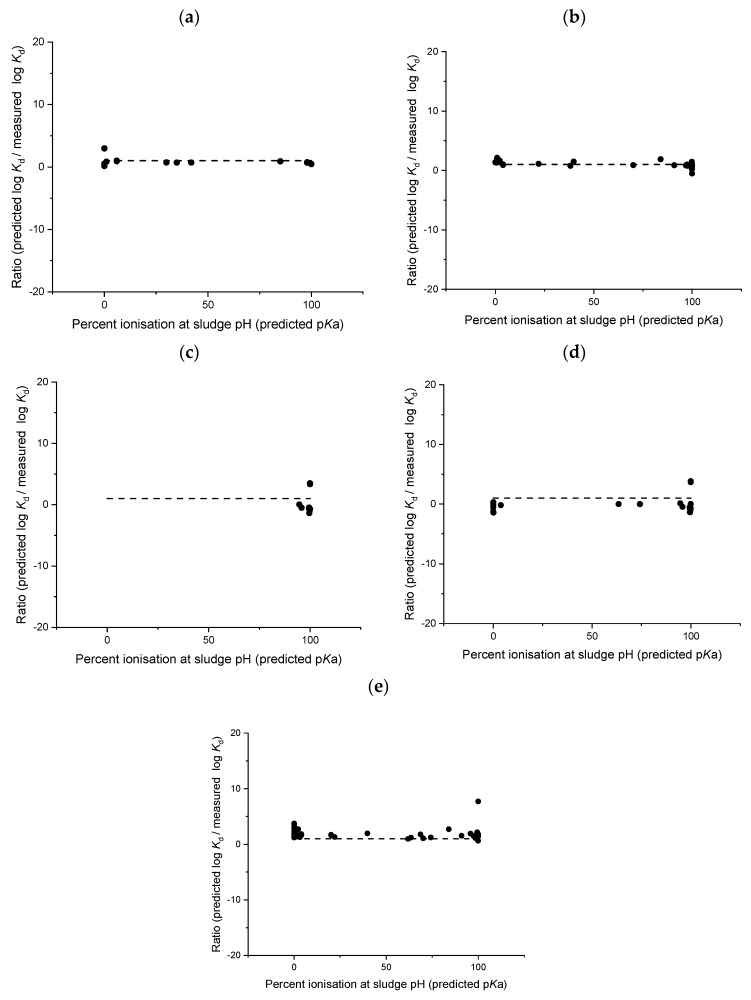
Ratio between predicted and measured sludge sorption coefficients and relationship to percent ionisation of test compound, Franco et al. [33] acids (**a**); Franco et al. [33] bases (**b**); Sathymoorthy and Ramsburg [32] ionised acids (**c**); Sathymoorthy and Ramsburg [32] acids (**d**); Berthod et al. [34] (**e**).

**Table 1 toxics-08-00013-t001:** Summary of soil sorption models selected for analysis.

Reference	Model	Specified Chemical Range of Applicability	Model Training Set
Bintein and Devillers [28] ^a^	Log *K*_d_ = 0.93 log *K_ow_* + 1.09 log *ƒ*oc + 0.32 CFa − 0.55 CFb’ + 0.25 Where: CFa = log (1(/1 + 10^pH−p*Ka*^)) CFb’ = log (1(/1 + 10^p*Ka−*(pH−2)^))	3.07 ≤ p*K*a ≤ 8.85 ^b^ 0.12 ≤ log *K*_ow_ ≤ 6.42 ^b^	Organic chemicals (not including pharmaceuticals) *(n* = 229*, r*^2^ = 0.96*)*
Sabljic et al. [37] (TGD)	Log *K*_OC_ *=* 0.10 + (0.81) log *K_ow_*	1 ≤ log *K*_ow_ ≤ 7.5	Hydrophobic chemicals (*n* = 81*, r*^2^ = 0.94*)*
Sabljic et al. [37] (TGD)	Log *K*_OC_ *=* 0.32 + (0.60) log *K_ow_*	1 ≤ log *K*_ow_ ≤ 7.5	Organic acids (*n* = 23*, r*^2^ = 0.87*)*
Kah and Brown [24]	Log *K*_d_ = 0.13 Log *D* + 1.02 Log OC − 1.51	1.97 ≤ p*K*a ≤ 4.94 ^b^ 1.2 ≤ log *K*_ow_ ≤ 4.3 ^b^	Ionisable pesticides *(n* = 90*, r*^2^ = 0.39*)*
Franco and Trapp [31]	Log *K*_OC_ = log (*ƒ*neutral 10^0.54^·^log*P*n+1.11^ + *ƒ*ion 10^0.11^·^log*P*n+1.54^)	0 < p*Ka* < 12 −2.18 < log *P*_n_ < 8.50	Organic acids (including 5 basic pharmaceuticals) *(n* = 62*, r*^2^ = 0.54*)*
Franco and Trapp [31]	Log *K*_OC_ = log (*ƒ*neutral 10^0.37^·^log*P*n+1.70^ + *ƒ*ion 10^p*Ka*0.65^·*^ƒ^*^0.14^)	2 < p*Ka* < 12 −1.66 < log *P*_n_ < 7.03	Organic bases (including 5 basic pharmaceuticals) *(n* = 43*, r*^2^ = 0.76*)*
Franco et al. [16]	*K*_OC_ = (10^0.54^·^log*P*n+1.11^)/(1 + 10^(pHsoil−0.6−p*Ka*)^) + (10^0.11^·^log*P*n+1.54^)/(1 + 10^(p*Ka*-pHsoil+0.6^)	Monovalent acids p*Ka* < 12 −2.18 < log *P*_n_ < 8.50	Organic acids *(r*^2^ = 0.70*)*
Droge and Goss [30]	*K*_d_ = *K*_CEC,CLAY_ CEC_CLAY_ + *f*_OC_ ·*D*_OC,IE_ = *K*_CEC,CLAY_ · (CEC_SOIL − 3.4 *f*OC_) + *f*_OC_ ·*D*_OC,IE_ Where: Log *K_CEC, CLAYS_ =* 1.22 (±0.15) Vx − 0.22 (±0.05) NAi + 2.09 (±0.05)	Strong bases (monovalent)	Organic cations (including pharmaceuticals) ^c^

^a^ Model presented in paper calculates Log *K*_p_. As this is defined as the soil sorption coefficient, which is what log *K*_d_ has been reported as in this analysis, Log *K*_p_ has been changed to log *K*_d_ for consistency; ^b^ Applicability domain not specified, calculated from information provided on physicochemical properties of training set chemicals; ^c^ Not a training set, model was developed using sorption data for a range of organic cations; log *K*_d_, (log) soil sorption coefficient; log *K*_OC_, (log) soil sorption coefficient normalised to organic carbon; log *K*_ow_, (log) octanol-water partition coefficient; p*K*a, negative log of the acid dissociation constant; *f*oc, fraction organic carbon in soil; CFa, concentration of anionic species in relation to pH; CFb’, protonated species concentration in relation to pH; *f*neutral/*f*ion, fraction of nonionised and ionic species, respectively calculated according to Franco and Trapp [31]; log *P*n and log *P*ion, (log) octanol–water partition coefficient of the nonionised molecule and of the ionic species, respectively, calculated according to Franco and Trapp [31]; log *D*, (log) lipophilicity corrected to soil pH; *K_CEC,CLAY_* and *D_OC,IE_* are the references sorption coefficient for the clay and organic content fraction in soil, respectively. Log *K_CEC,CLAY_* = 1.22 Vx − 0.22NAi + 1.09; log *K_CEC,CLAY_* = 1.53Vx + 0.32NAi − 0.27. Vx, molecular volume (L mol^−1^); NAi, number of hydrogens bound by the charged nitrogen.

**Table 2 toxics-08-00013-t002:** Summary of sludge sorption models selected for analysis.

Reference	Model	Specified Range of Chemical Applicability	Model Training Set
Franco et al. [33]	*K*_OC_ = *ƒ*n 10^0.54·log*K*own + 1.11^ + *ƒ*ion10^0.11·log*K*own + 1.54^	Monovalent acids p*Ka* < 10	
Franco et al. [33]	*K*_OC_ base = 10^0.31·log *D* + 2.78^	Monovalent bases p*Ka* > 4	
Sathyamoorthy and Ramsburg [32]	Log *K*_d_ *=* [5.88 ± 1.69] + [(0.37 ± 0.05)log *D*] + [(0.30 ± 0.05)nHBA] + [(–3.56 ± 0.78)log MV]	^a^	Negatively charged pharmaceuticals *(n* = 44*, r*^2^ = 0.60*)*
Sathyamoorthy and Ramsburg [32]	Log *K*_d_ *=* (4.54 ± 1.36) + [(0.39 ± 0.04)log *D*] + [(0.32 ± 0.04) nHBA] + [(–2.41 ± 0.59)log MV] + [(−0.86 ± 0.25)log TPSA	^a^	Negatively charged and uncharged pharmaceuticals *(n* = 109*, r*^2^ = 0.64*)*
Berthod et al. [34]	Artificial Neural Network (ANN)	–4.55 ≤ log *K*_ow_ ≤ 7.05 ^b^	Ionisable pharmaceuticals

^a^ Range not specified and unable to determine applicability domain from data provided in paper; ^b^ Applicability domain not specified, calculated from information provided on physicochemical properties of training set chemicals; log *D*, (log) pH corrected octanol water partition coefficient; *K*_OC_, soil sorption coefficient normalised to organic carbon; log *K*_ow_, (log) octanol-water partition coefficient; p*K*a, negative log of the acid dissociation constant; log *K*_d_, (log) soil sorption coefficient; *f*neutral/*f*ion, fraction of nonionised and ionic species, respectively, calculated according to Franco and Trapp [31]; nHBA, number of hydrogen bond acceptors; MV, molecular volume (Å^3^); TPSA, topological polar surface area (Å^2^).

**Table 3 toxics-08-00013-t003:** Assessment of the performance of previously published soil sorption models in their ability to predict soil sorption coefficients (log *K*_d_).

Reference	Charge Group Relevant to Model	Model Performance					Number of Data within Model Applicability Domain
		*r* ^2^	NSE	RMSE	RMSE/MAE	% within a Factor of 10	
Bintein and Devillers [28]	Acids	0.005	−4.43	0.39	0.18	26	38 ^a^
Bintein and Devillers [28]	Bases	0.08	−9.65	0.32	0.12	24	85 ^a^
Sabljic et al. [37] (TGD) ^b^	Hydrophobic chemicals	0.07	−1.54	0.62	0.51	55	194 ^c^
Sabljic et al. [37] (TGD) ^b^	Acids	0.04	−1.25	0.62	0.50	48	77
Kah and Brown [24]	Acids	0.003	−7.71	0.91	0.97	71	7 ^a^
Franco and Trapp [31] ^b^	Acids	0.17	−0.26	0.70	0.67	68	68
Franco and Trapp [31] ^b^	Bases	0.07	−0.31	0.65	0.58	55	114
Franco and Trapp [16] ^b^	Acids	0.17	−0.26	0.70	0.67	68	68
Droge and Goss [30]	Bases	0.29	0.18	0.79	0.93	71	66

^a^ Defined applicability domain based on information provided for training set in original paper; ^b^ Predicted log *K*_oc_ were converted to log *K*_d_ using information on test soil organic carbon content (*K*_d_ = *K*_OC_ 0.01 OC(%)); ^c^ Organic carbon content not available for all datasets. Conversion from log *K*_oc_ to log *K*_d_ was, therefore, not possible for some chemicals.

**Table 4 toxics-08-00013-t004:** Assessment of the performance of previously published sludge sorption models in their ability to predict sludge sorption coefficients.

Reference	Charge Group Relevant to Model	Model Performance					Number of Data within Model Applicability Domain
		*r* ^2^	NSE	RMSE	RMSE/MAE	% within a Factor of 10	
Franco et al. [33] ^a^	Acids	0.07	0.99	0.89	0.94	60	15 ^b^
Franco et al. [33] ^a^	Bases	0.04	−0.07	0.76	0.90	67	28
Sathyamoorthy and Ramsburg [32]	Acids (with a negative charge at sludge pH)	0.08	−12.68	0.31	0.10	11	10
Sathyamoorthy and Ramsburg [32]	Acids	0.04	−9.28	0.32	0.11	4	23
Berthod et al. [34]	All	0.21	−2.38	0.54	0.32	21	66 ^c^
Berthod et al. [34]	Acids	0.28	−4.42	0.56	0.34	19	21
Berthod et al. [34]	Bases	0.21	−1.76	0.52	0.30	19	32
Berthod et al. [34]	Multiple ionisable groups	0.01	−17.50	0.52	0.33	30	13

^a^ Predicted *K*_oc_ values were converted to *K*_d_ using information on test sludge organic carbon content; ^b^ Organic carbon content not available for all datasets. Conversion from log *K*_oc_ to log *K*_d_ was, therefore, not possible for some chemicals (*K*_d_ = *K*_OC_ 0.01 OC(%)); ^c^ 18 chemicals included in the iPiE database were used to develop the model and, therefore, were removed from the analysis of this model.

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
