# Peer review of "Evaluation of Existing Models to Estimate Sorption Coefficients for Ionisable Pharmaceuticals in Soils and Sludge"

_toxics, 2020, doi:10.3390/toxics8010013_

Round 1

Reviewer 1 Report

The manuscript is well written and findings contribute to overall knowledge in this field.

I have few remarks:

Table 1

Model Binetin and Devillers [34]a – I think that brackets are missing (1/1 is simply 1)

CFa = log (1/(1+10pH-pKa))

CFb’ = log (1/(1+10pKa-(pH-2)))

Franco and Trapp [28] (second line): 2<pKa>12 …. this does not have a sense.

Log KOC = log (ƒneutral · 100.37 · log Pn + 1.70 + ƒion · 10pKa 0.65 · ƒ· 0.14) …. I suppose that “fneutral” and “fion” are fractions of neutral and ionic molecules, respectively. (By the way I did not find this explanation in the text. In Table 2 only “fn” is used.) What is “f”?

Line  163 …(L/mol)…

References

Perhaps format of references can be unified. For instance:

13. Rybacka A, Andersson PL. 2016. Considering ionic state in modeling sorption of pharmaceuticals to sewage 566 sludge. Chemosphere. doi:10.1016/j.chemosphere.2016.09.014. …. Instead of DOI No. of issue and pages can be included as for some other citations. Similarly, 18, 19, 20, 35, 41, 42, 49…..

Supplemental materials

What is the reason that names of compounds are not included in both tables? I understand that for general applications of the models for different functional groups this information seems to be unimportant. However, it would be nice to know, which compounds were actually included in this study.

Author Response

We thank the reviewer for their comments. We have included our reply, addressing each comment, in the attached Word document.

Reviewer 2 Report

This work evaluated the prediction accuracy of sorption in soils and sludge for ionisable pharmaceuticals to use many kinds of present models. The results are interesting and helpful for assess the environmental risk of a pharmaceutical. However, I can find a lot of mistakes. I recommend that authors should revise them. The manuscript can be accepted after revision. The modified points are shown as follows.

1) line.149, Table 1

The format of table should be corrected. Six bold lines in Table 1 should be changed to normal lines.

2) line.150

Log Kp is changed to log Kp.

3) line.151

log Kd, Log Kp, and log Kd should be changed to log Kd, log Kp, log Kd, respectively.

4) line.154

log Kd and log KOC should be changed to log Kd and log KOC, respectively.

5) line.162

Log KCEC,Clay should be changed to log KCEC,CLAY.

6) line.165

“soil sorption” should be changed to “sludge sorption”.

7) line.267, Figure 2

It is difficult to see Figure 2. Compact figure is better. How about the style of two-line and three-row?

8) line.268-269

“pharmaceuticals Binetin and Devillers (a), Sabljic et al. (b), Franco et al. 2008 (c), Franco et al., 2009 268 (d), Kah and Brown (e).” should be changed to “pharmaceuticals, Binetin and Devillers [34] (a), Sabljic et al. [35] (b), Franco et al. [28] (c), Franco et al. [16] (d), Kah and Brown [21] (e).”

9) line.283, Figure 3

It is difficult to see Figure 3. How about the style of one-line and three-row?

10) line.284

“pharmaceuticals Binetin and Devillers (a), Franco et al. 2008 (b), Droge and Goss (c).” should be changed to “pharmaceuticals, Binetin and Devillers [34] (a), Franco et al. [28] (b), Droge and Goss [27] (c).”

11) line.340, Figure 4

It is difficult to see Figure 4. How about the style of two-line and three-row?

12) line.340-342

“Franco et al. 2013 – Acids (1), Franco et al. 2013 – Bases (2), Sathymoorthy and Ramsburg – ionised Acids (3), Sathymoorthy and Ramsburg – Acids (4), Berthod et al. 2017 (5).” should be changed to “Franco et al. [30] – Acids (1), Franco et al. [30] – Bases (2), Sathymoorthy and Ramsburg [29] – ionised Acids (3), Sathymoorthy and Ramsburg [29] – Acids (4), Berthod et al. [31] (5).”

13) line.364, Figure 5

It is difficult to see Figure 5. How about the style of three-line and three-row?

14) line.365-367

“Binetin and Devillers – Acids (a), Binetin and Devillers – Bases (b), Sabljic et al. – Acids (c), Sabljic et al. – Hydrophobics (d), Franco et al. 2008 – Acids (e), Franco et al. 2008 – Bases (f), Franco et al., 2009 (g), Kah and Brown (h), Droge and Goss (i).” should be changed to “Binetin and Devillers [34] – Acids (a), Binetin and Devillers [34] – Bases (b), Sabljic et al. [35] – Acids (c), Sabljic et al. [35] – Hydrophobics (d), Franco et al. [28] – Acids (e), Franco et al. [28] – Bases (f), Franco et al. [16] (g), Kah and Brown [21] (h), Droge and Goss [27] (i).”

15) line.385, Figure 6

It is difficult to see Figure 6. How about the style of two-line and three-row?

“soil pH” in the horizontal axis should be changed to “sludge pH”.

16) line.385-387

“Figure 6. Ratio between predicted and measured soil sorption coefficients and relationship to percent ionisation of test compound: Franco et al. 2013 – Acids (a), Franco et al. 2013 – Bases (b), Sathymoorthy and Ramsburg – ionised Acids (c), Sathymoorthy and Ramsburg – Acids (d), Berthod et al. 2017 (e).” should be changed to “Figure 6. Ratio between predicted and measured sludge sorption coefficients and relationship to percent ionisation of test compound: Franco et al. [30] – Acids (a), Franco et al. [30] – Bases (b), Sathymoorthy and Ramsburg [29] – ionised Acids (c), Sathymoorthy and Ramsburg [29] – Acids (d), Berthod et al. [31] (e).”

Author Response

(The authors gave the same response as above.)

Reviewer 3 Report

This manuscript by Carter et al. aims to assess several models employed for the estimation of the solid water partition coefficient of pharmaceuticals onto soils and sludges.

First of all, the writing of the manuscript is very sound and pleasant to follow, even if in my view several references are missing. The main results of this manuscript are quite uncommon, as finally, none model correctly fits with the data, but this work clearly deserves for publication. However, there are two points that have to be adressed prior publication. Hence, i recommend the publication of this work after moderate modifications. Specific comments are given below.

In the introduction section, i think that you should add some references, especially in the section dealing with the adsorption mechanism of pharmaceuticals with each soil compartments which is not enoughly detailed.

Table S1 and S2: There is several errors to fix in this Table. First of all, please indicate the meaning of N/A and NR in the caption. Concerning zwitterionic pharmaceuticals, why the ionised % is always 0? Based on the pKa of the base and the acid, you can determine the %ionised?

During adsorption experiments, were the starting concentrations constant for each compound, the solid/liquid ratio? I cannot find this information, and because you work on Log Kd values, if these two parameters were not constant, it would question the validity of your dataset. Also, was the background solution constant as well? You should provide such information at least in supporting information.

In general throughout the manuscript, you reason in term of "acid", "base" and "zwitterion". Yet, for some data with a %ionised of 0 (especially for acids and bases") wouldn't it be preferable to speak about "neutral" or "uncharged" chemical?

Following the previous comment, you should clarify the used dataset for each model. You provide the number of data within applicability domain in Table 3, but it needs to be clearer.

L485-486 "are highly protonated" (> 90% ionised) the protonation did not ionised acidic compounds, please revise or reword.

Conclusions: In my view, you should be more precise in your conclusions. You should clearly demonstrate the added value of your study on assessing such models and avoid any vague sentence as "More work is therefore needed", what kind of works? What did you identify as problematic?

Author Response

(The authors gave the same response as above.)

Round 2

Reviewer 3 Report

The authors have performed the recommended modifications, and their manuscript is now suitable for publication.

Just for clarification on one point of the first revision,

"L485-486 "are highly protonated" (> 90% ionised) the protonation did not ionised acidic compounds, please revise or reword.

The authors are unsure what this comment is in relation to. If the reviewer is referring to L. 468 and the sentence “when a chemical is highly charged (> 90% ionised) this also results in a larger ratio between predicted and measured log Kd values (Figure 5).” We believe this statement to be true. This is in relation to the ratio of predicted and measured Kd values in the soil Figure (Figure 5), whereby when you have a large percentage of ionisation (or charge) this results in a larger ratio between the predicted and measured Kd values. The same is also true for the sludge data (Figure 6) however this is to a lesser extent."

The meaning of my comment was that the protonation does not systematically generate ionization, as for example in case of acids. Could you clarify this sentence at proof stage ?